# Radiomic Analysis Based on Magnetic Resonance Imaging for Predicting PD-L2 Expression in Hepatocellular Carcinoma

**DOI:** 10.3390/cancers15020365

**Published:** 2023-01-05

**Authors:** Yun-Yun Tao, Yue Shi, Xue-Qin Gong, Li Li, Zu-Mao Li, Lin Yang, Xiao-Ming Zhang

**Affiliations:** 1Medical Imaging Key Laboratory of Sichuan Province, Interventional Medical Center, Department of Radiology, Medical Research Center, Affiliated Hospital of North Sichuan Medical College, Nanchong 637000, China; 2Department of Pathology, Affiliated Hospital of North Sichuan Medical College, Nanchong 637000, China

**Keywords:** PD-L2, MRI, hepatocellular carcinoma, radiomics, immunotherapy target, immune checkpoint blockade

## Abstract

**Simple Summary:**

Immunotherapy targeting the programmed death-1 (PD-1)/programmed death ligand-1 (PD-L1) has attracted worldwide attention and is setting off a revolution in cancer treatment, bringing new hope to cancer patients. PD-L2 is another ligand of PD-1 and a promising immunotherapy marker. This study aimed to find a prediction model based on the radiomic characteristics of magnetic resonance images to noninvasively predict the expression of PD-L2 in liver cancer before surgery, thereby to provide a reference for the choice of immune checkpoint blockade therapy.

**Abstract:**

Hepatocellular carcinoma (HCC) is the sixth most common malignant tumour and the third leading cause of cancer death in the world. The emerging field of radiomics involves extracting many clinical image features that cannot be recognized by the human eye to provide information for precise treatment decision making. Radiomics has shown its importance in HCC identification, histological grading, microvascular invasion (MVI) status, treatment response, and prognosis, but there is no report on the preoperative prediction of programmed death ligand-2 (PD-L2) expression in HCC. The purpose of this study was to investigate the value of MRI radiomic features for the non-invasive prediction of immunotherapy target PD-L2 expression in hepatocellular carcinoma (HCC). A total of 108 patients with HCC confirmed by pathology were retrospectively analysed. Immunohistochemical analysis was used to evaluate the expression level of PD-L2. 3D-Slicer software was used to manually delineate volumes of interest (VOIs) and extract radiomic features on preoperative T2-weighted, arterial-phase, and portal venous-phase MR images. Least absolute shrinkage and selection operator (LASSO) was performed to find the best radiomic features. Multivariable logistic regression models were constructed and validated using fivefold cross-validation. The area under the receiver characteristic curve (AUC) was used to evaluate the predictive performance of each model. The results show that among the 108 cases of HCC, 50 cases had high PD-L2 expression, and 58 cases had low PD-L2 expression. Radiomic features correlated with PD-L2 expression. The T2-weighted, arterial-phase, and portal venous-phase and combined MRI radiomics models showed AUCs of 0.789 (95% CI: 0.702–0.875), 0.727 (95% CI: 0.632–0.823), 0.770 (95% CI: 0.682–0.875), and 0.871 (95% CI: 0.803–0.939), respectively. The combined model showed the best performance. The results of this study suggest that prediction based on the radiomic characteristics of MRI could noninvasively predict the expression of PD-L2 in HCC before surgery and provide a reference for the selection of immune checkpoint blockade therapy.

## 1. Introduction

Hepatocellular carcinoma (HCC) is the sixth most common malignant tumour and the third leading cause of cancer death in the world [1]. Although the treatment of HCC has made great progress, the long-term efficacy is still not satisfactory [2,3,4]. Therefore, it is necessary to seek new treatment methods for clinical practice. Tumour immunotherapy, one of the most promising cancer treatment methods, regulates the immune system and enhances antitumour immunity, thereby achieving the goal of inhibiting and killing tumour cells [5].

Programmed death-1 (PD-1) is an important immunosuppressive molecule in the human body. It binds to programmed death ligand-1 (PD-L1) to inhibit the activation of T cells; it allows tumour cells to achieve immune escape, resulting in poor prognosis [6,7]. In recent years, tumour immunotherapy targeting PD-1 and PD-L1 has attracted much attention. Some patients with negative PD-L1 expression still respond to PD-1 checkpoint inhibitors, suggesting that other PD-1 ligands may underlie the clinical response to these treatments [8]. PD-L2 is another ligand of PD-1 that can be induced in tumours by interferon exposure, leading to immune evasion [9,10]. PD-L2 expression mostly associated with PD-L1 expression, can also occur in the absence of PD-L1 [8]. Previous studies showed that PD-L2 is expressed in pancreatic ductal adenocarcinoma, non-small-cell lung cancer, glioma, and HCC and high PD-L2 expression is associated with poor prognosis [8,11,12,13,14,15,16]. A study reported that PD-L2 expression is upregulated in tumour-associated macrophages (TAM), and its inhibitory effects become evident when PD-L1 function is abrogated by anti-PD-L1 monoclonal antibody (mAb) [17]. These studies suggested that, although PD-L1/PD-1-dependent suppression is the primary mechanism of immune evasion in cancer, alternative mechanism of PD-L2 upregulation, may compensate once PD-L1 function is dampened [17]; PD-L2 expression may provide information beyond that of PD-L1 in predicting clinical results to targeted immunotherapy, and is a promising marker to anti-PD-1 targeted agents [8,11,12,18].

It is important to accurately predict the immune status of cancer patients before immunotherapy [19]. The emerging radiomics extracts many potential image features that cannot be recognized by the human eyes based on the existing images of imaging such as magnetic resonance imaging (MRI), computed tomography (CT), and ultrasound to provide a reference for precise treatment decision making [20]. Some scholars have reported the application value of radiomics to HCC differentiation, histological grading, microvascular invasion (MVI) status, treatment response, and prognostic prediction [21,22,23,24,25,26,27]. To date, no radiomics model has been reported to predict the expression of PD-L2 in HCC before surgery. This study investigated the efficacy of the preoperative prediction of PD-L2 expression in HCC by a new MRI radiomics model.

## 2. Materials and Methods

### 2.1. Patient

This study included 108 patients with HCC who underwent surgical resection in our hospital between January 2018 and June 2021. Inclusion criteria: 1. HCC was confirmed by surgical resection and pathology. 2. MRI examination was performed within 2 weeks before treatment. 3. No other antitumour treatment was received before surgery. 4. The patient’s clinical data were completed. Exclusion criteria: 1. The maximum diameter of the lesion was <10 mm. 2. The image quality was poor (Figure 1).

### 2.2. Immunohistochemistry

Immunohistochemical staining was performed to detect the expression of PD-L2 in HCC tissues. The staining results were scored as described [13]. Two pathologists who were unaware of the clinical results scored images based on the percentage of positively stained cells and the intensity of staining. Positive cells were those with yellow or brown cell membranes and/or cytoplasm. Scoring criteria for the percentage of positive cells: <10%: 0 points; 10–50%: 1 point; >50%: 2 points. Scoring criteria for staining intensity: no staining: 0 points; weak staining: 1 point; strong staining: 2 points. The total score was the sum of the above two. According to the total score, HCC patients were divided into two groups: low PD-L2 expression (<3 points) and high PD-L2 expression (≥3 points). For controversial results, the final score was reached through discussion by the two radiologists.

### 2.3. MR Image Acquisition

MRI scans were performed using the Discovery 750 3.0-T superconducting magnetic resonance imaging scanner (GE HealthCare, Chicago, IL, USA), with a 32-channel phased-array surface coil. The scan sequences included T1-weighted imaging (T1WI), fat-suppressed T2-weighted imaging (FS-T2WI), and dynamic contrast-enhanced (DCE) MRI (Table 1). Dynamic enhanced scanning: A high-pressure syringe was used to inject 15–20 mL of the contrast agent gadolinium–diethylenetriaminepentaacetic acid (Shanghai Bracco Sine Pharmaceutical Co., Ltd.) through the superficial vein on the back of the hand at a speed of 2–2.5 mL/s. The images of the hepatic arterial and portal venous phases were acquired.

### 2.4. Tumour Segmentation and Feature Extraction

The MR images of the study subjects were downloaded and exported through PACS and imported into 3D-Slicer software. A radiologist with 7 years of work experience manually delineated the volume of interest (VOI) on the FS-T2WI sequence, DCE-MRI arterial phase, and portal venous phase sequence [28,29] (Figure 2). When sketching the VOI, attention should be given to avoid the surrounding bile ducts and blood vessels. The original images were passed through the Laplacian of Gaussian filter and the wavelet filter [30] to obtain the derived images for each patient. The extracted features were derived from the original images and the derived images. A total of 1130 features were extracted from each sequence. The arterial phase and portal vein phase feature sets of FS-T2WI and DCE-MRI, and the combined feature set of the former three were established [31]. The extracted features were standardized using R software to eliminate the unit limit of each feature data [32]. Two observers randomly selected 50 HCC patients to delineate the VOIs of FS-T2WI sequence, arterial-phase, and portal venous-phase images and extract the radiomic features. The interclass correlation coefficient (ICC) was calculated to evaluate interobserver consistency. Features with ICC < 0.75 were excluded [33].

### 2.5. Feature Screening and Model Establishment

The least absolute shrinkage and selection operator (LASSO) and stepwise regression analysis were used to select the optimal radiomic feature [34,35]. A logistic regression model was established for the selected optimal feature set. A 5-fold cross-validation method was used to verify the performance of the model [36]. The predictive performance of each model was evaluated by the area under the receiver operating characteristic (ROC) curve (AUC).

### 2.6. Statistical Analysis

R software (version 4.1.1, https://www.r-project.org/, accessed on 12 August 2021) was used for statistical analysis. Categorical data are expressed as whole numbers and proportions, and they were compared between groups by the chi-squared test. LASSO was performed using the “glmnet” software package of the R statistical environment. Statistical results were considered significant when *p* < 0.05.

## 3. Results

A total of 108 HCC patients were enrolled in this study, aged 23–73 years (mean: 53 years); 95 (88%) were male, 81 (75%) had liver cirrhosis, and the longest tumour diameter ranged from 2.0 cm to 20.1 cm (the average longest diameter was 6.4 cm). There were 50 patients in the high-PD-L2-expression group and 58 patients in the low-PD-L2-expression group (Table 2, Figure 3).

Finally, 10, 8, 10, and 14 optimal features were obtained from the FS-T2WI, enhanced-scan arterial-phase, portal venous-phase, and combined datasets, respectively, which were used to establish the prediction model. The AUCs of the FS-T2WI sequence, contrast-enhanced arterial-phase, and portal venous-phase sequence models and the combined model in the training set were 0.852 (95% CI: 0.781–0.924), 0.814 (95% CI: 0.734–0.895), and 0.857 (95% CI: 0.734–0.895), and 0.955 (95% CI: 0.921–0.989), respectively. The AUCs in the validation set were 0.789 (95% CI: 0.702–0.875), 0.727 (95% CI: 0.632–0.823), 0.770 (95% CI: 0.682–0.857), and 0.871 (95% CI: 0.803–0.939), respectively (Figure 4, Table 3).

There was no significant difference in the efficacy of PD-L2 expression in HCC patients evaluated by the single-sequence models (*p* > 0.05), while the AUCs of the training set and validation set of the combined model were significantly higher than that of any of the above single-sequence model (*p* < 0.05) (Figure 5).

## 4. Discussion

MRI-based radiomic features can be used as a non-invasive predictor of immune characteristics of liver cancer, which may be helpful for the treatment stratification of liver cancer patients [37,38,39]. Hectors [38] showed that MRI-based textural features were significantly correlated with PD-L1 expression. Gu [39] established a support vector machine model based on MR imaging radiomic features for predicting Glypican 3 expression in HCC patients and achieved good results. The AUC of the training group was 0.879, and the AUC of the validation group was 0.871. When it was combined with the clinical risk predictor AFP, its predictive performance was significantly improved. Its AUC was 0.926 in the training group and 0.914 in the validation group.

As one of the ligands of PD-1, PD-L2 is involved in the composition of the immune microenvironment of HCC patients and is associated with poor prognosis [13]. The use of non-invasive methods to predict the expression of PD-L2 is conducive to the development of individualized liver cancer protocols. The results of this study showed that the logistic regression model established based on the radiomic characteristics of the FS-T2WI sequence, the contrast-enhanced arterial phase sequence, and the portal venous phase sequence could well predict PD-L2 expression in HCC patients. The combined model had significantly improved predictive performance, possibly because the MRI combined model was constructed from data from different sequences, and their different information was complementary to each other and could more fully and accurately reflect the internal characteristics of the tumour. This conclusion is consistent with the literature [21,23,40].

Different MRI sequences provide different information, and their contributions to the combined model are also different. Wang [41] established an MRI-based radiomics model to predict the 5-year survival rate of HCC patients, the contribution of different sequences to the prediction model was as follows: DCEI > DWI > T2WI > T1WI. Among the 14 features of the combined model established in this study, there were six features from the FS-T2WI and eight features from the enhanced sequence (three in the arterial phase and five in the portal venous phase), which is consistent with the literature [42].

The normal liver is mainly supplied by the portal vein; HCC is mainly supplied by the hepatic artery. Dynamic-enhanced MRI can display the characteristics of dynamic-enhanced images of HCC and provide information on differences in vascular distribution. Zhang et al. showed that the radiomics model established based on the contrast-enhanced sequence was more effective at predicting the MVI status and early recurrence of HCC than other sequences [24,43]. In Zhao’s [44] study on preoperative prediction of early recurrence after partial resection of HCC, the AUC of their MRI radiomics model based on the portal vein phase was 0.750, which was superior to that of the T2WI model (AUC: 0.710) and the arterial-phase model (AUC: 0.717).The radiomics model based on multiparametric MRI radiomics features presented the best performance (AUC: 0.831) among all radiomics models in the training cohort. The combined nomogram (AUC: 0.873) incorporated clinicopathologic radiologic (CPR) factors (MVI, pathological grading, and tumour size) outperformed the radiomics model and the CPR model. These results described above may be related to the following factors [44,45]: (1) MVI is one of the important prognostic factors and correlated closely with early recurrence of HCC after surgical treatment. (2) Due to the large signal differences caused by the washout of the typical enhancement method of HCC, the tumour boundary on the portal vein phase in the dynamic enhanced image can be clear so that the preoperative portal vein phase image can highlight the heterogeneity of HCC. (3) Multiparametric MRI contains more potential tumour heterogeneity information.

Discrete wavelet transform with eight filters (four high-pass and four low-pass) [46] can help identify sudden changes in details or intensity in an image [30]. After the feature screening in this study, the wavelet features in each sequence model and the combined model were the most preserved, in line with previous studies [47,48,49,50]. Yuan [49] established a nomograph based on enhanced CT images and clinical factors to predict the efficacy of anti-PD-1 treatment in patients with advanced HCC, and the features extracted after wavelet filter transformation accounted for 8/9 (88.9%) of the features. They suggested that wavelet transform is a mathematical technique that can decompose special patterns hidden in a large amount of data. Xu [50] developed and validated a predictive model for the preoperative lymph node status in patients with intrahepatic cholangiocarcinoma, and the features used were all from wavelet features.

This study has the following limitations. (1) First, the manual segmentation method was used. Manual segmentation has higher accuracy than semiautomatic segmentation and automatic segmentation, so it is often used as the standard basis for evaluating other segmentation methods [28,29]. However, the problem of inaccurate segmentation caused by unclear boundaries of some images cannot be completely avoided. (2) The main purpose of this study was to investigate the performance of the MRI radiomics model in predicting the expression of PD-L2 in HCC patients, and only a logistic regression model was used. Other classification models, such as random forest and support vector machine, were not tried. The advantages and disadvantages of different classifiers were not compared. (3) This study was a retrospective study using single-centre data and the same MRI model. It lacked effective external validation, limiting the generalizability of our results to other centres.

## 5. Conclusions

This MRI-based radiomics model can predict the expression of PD-L2 in HCC patients before surgery and provide reference information for immune checkpoint blockade therapy.

## Figures and Tables

**Figure 1 cancers-15-00365-f001:**
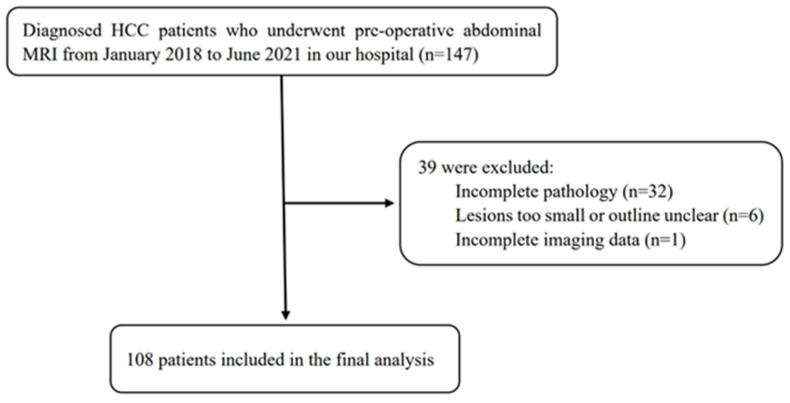
Flow chart of the study population.

**Figure 2 cancers-15-00365-f002:**
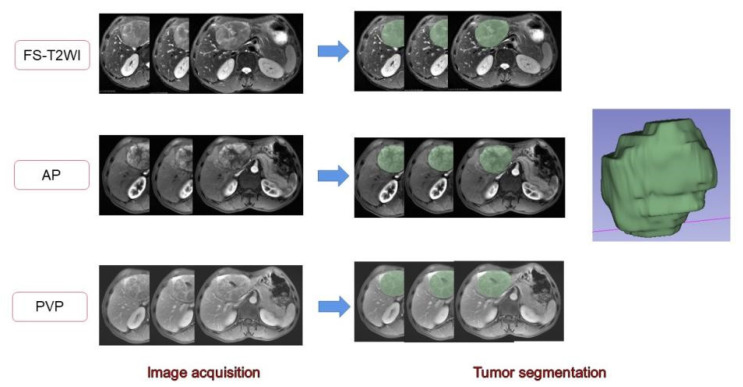
Delineation of the target area and generation of the full VOI. The 2D region of interests (ROIs) were manually delineated on FS-T2WI and dynamic contrast-enhanced MRI. Three-dimensional views were automatically generated for all ROIs.

**Figure 3 cancers-15-00365-f003:**
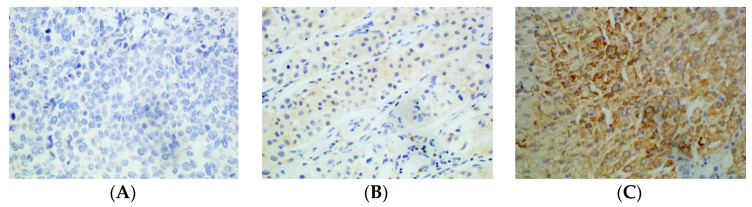
Immunohistochemical staining of HCC tissue using anti-PD-L2 antibody. (**A**): Negative. (**B**): Weakly positive. (**C**): Strongly positive (×400).

**Figure 4 cancers-15-00365-f004:**
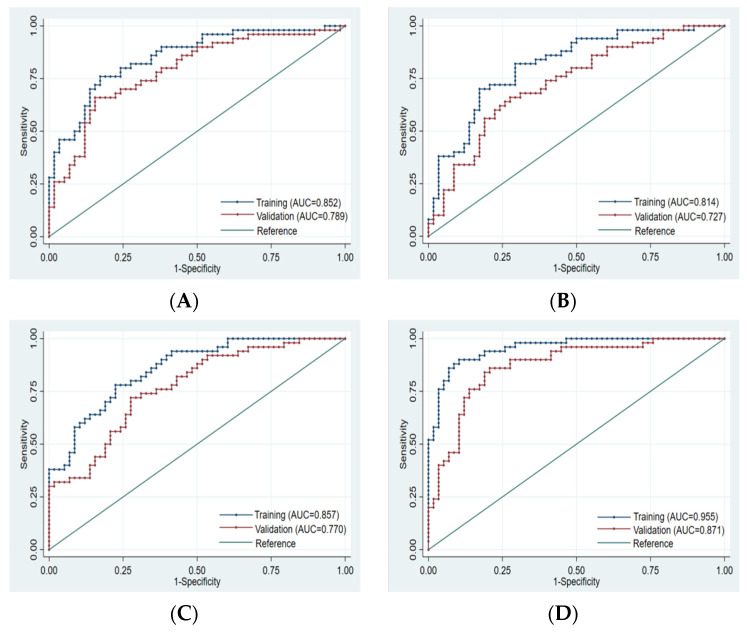
ROC curve of training set and validation set for each model. (**A**): FS-T2WI. (**B**): Arterial phase. (**C**): Portal venous phase. (**D**): Combined sequence. The blue curve represents the training set, and the red curve represents the validation set.

**Figure 5 cancers-15-00365-f005:**
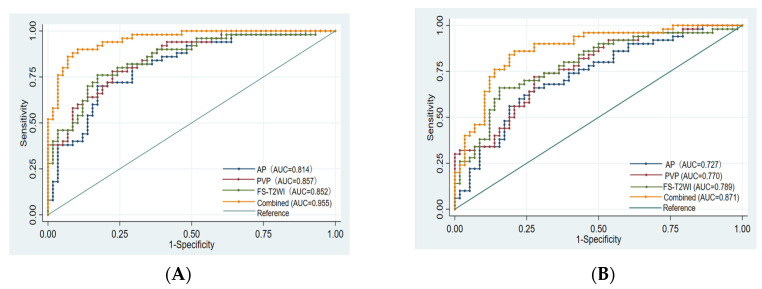
Comparison of the ROC curves for the prediction of PD-L2 expression by various models. (**A**): ROC of each model training set. (**B**): ROC of each model validation set. AUC: area under the ROC curve; AP: arterial phase; PVP: portal venous phase; Combined: combined model.

**Table 1 cancers-15-00365-t001:** MRI sequences and their related parameters.

Sequence	TR/TE (ms)	FA (°)	FOV (mm^2^)	ST (mm)	Matrix (mm^2^)
T1WI	4/2	12	320 × 320–360 × 360	2.5	260 × 192
FS-T2WI	2609/97	110	320 × 320–380 × 380	6	384 × 384
DCE-MRI	4/2	12	320 × 320–360 × 360	5	224 × 192

Note: T1WI:T1-weighted imaging: T1WI was acquired using a three-dimensional liver acquisition with volume acceleration (3D-LAVA) with axial slicing; FS-T2WI: axial T2-weighted imaging with fat suppression; DCE-MRI: dynamic contrast-enhanced MRI; TR: repetition time; TE: echo time; FA: flip angle; FOV: field of view; ST: slice thickness.

**Table 2 cancers-15-00365-t002:** Clinical characteristics of patients in the high and low PD-L2 expression groups.

Clinical Variables	Total (*n* = 108)	High PD-L2 Expression (*n* = 50)	Low PD-L2 Expression (*n* = 58)	*p*
Age (years)				0.105
≤60	69 (64%)	36 (72%)	33 (57%)	
>60	39 (36%)	14 (28%)	25 (43%)	
Sex (%)				0.163
Male	94 (87%)	46 (92%)	48 (79%)	
Female	14 (13%)	4 (8%)	10 (21%)	
AFP (ng/mL)				0.375
<20	37 (34%)	18 (36%)	19 (33%)	
20–400	25 (23%)	14 (28%)	11 (19%)	
≥400	46 (43%)	18 (36%)	28 (48%)	
Diameter (cm)				0.629
0–5	47 (44%)	23 (46%)	24 (41%)	
≥5	61 (56%)	27 (54%)	34 (59%)	
Hepatitis B				0.785
No	12 (11%)	7 (14%)	5 (9%)	
Yes	96 (89%)	43 (86%)	53 (91%)	
Liver cirrhosis				0.504
No	27 (25%)	14 (28%)	13 (22%)	
Yes	81 (75%)	36 (72%)	45 (78%)	
Portal vein tumour thrombus				0.664
No	82 (76%)	37 (74%)	45 (78%)	
Yes	26 (24%)	13 (26%)	13 (22%)	

Note: AFP: alpha-fetoprotein.

**Table 3 cancers-15-00365-t003:** Predictive performance of each model.

Model	AUC of Training Set (95% CI)	AUC of Validation Set (95% CI)	Accuracy	Sensitivity	Specificity	*p* Value
FS-T2WI	0.852 (0.781–0.924)	0.789 (0.702–0.875)	73.15%	66.00%	79.31%	0.0051
AP	0.814 (0.734–0.895)	0.727 (0.632–0.823)	69.44%	62.00%	75.86%	0.0006
PVP	0.857 (0.789–0.925)	0.770 (0.682–0.857)	71.30%	70.00%	72.41%	0.0018
Combined	0.955 (0.921–0.989)	0.871 (0.803–0.939)	82.41%	86.00%	79.31%	Reference

Note: AUC: area under the ROC curve; CI: confidence interval; FS-T2WI: fat suppression T2-weighted imaging; AP: arterial phase; PVP: portal venous phase; Combined: FS-T2WI + AP + PVP. *p* value versus combined.

## Data Availability

The data presented in this study is available on request from the corresponding author.

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
