# Peer review of "Radiomic Analysis Based on Magnetic Resonance Imaging for Predicting PD-L2 Expression in Hepatocellular Carcinoma"

_cancers, 2023, doi:10.3390/cancers15020365_

Round 1

Reviewer 1 Report

This is a timely article and presents interesting results for prediction of PDL2 experession in HCC. 

Abstract: Concise

Introduction: Well written

Materials and Methods: Well described

Results: Easily comprehsible.

Discussion: Please rephare sentence line 215-218 (The reason may be that MVI can change tumour perfusion through micro-portal vein infiltration,  thereby detecting the difference in contrast enhancement between HCC with MVI and HCC without MVI, while the early recurrence of HCC was significantly correlated with  MVI). I could not understand what the authors meant.

Limitations are described

Author Response

Re: Dear Reviewer, thank you all for handling the review of our manuscript. Based on your highly constructive comments and suggestions, we have addressed all of your comments and suggestions. We have revised the sentence line 215-218.

The modifications are as follows:

In Zhao’s [41] study on preoperative prediction of early recurrence after partial resection of HCC, the AUC of their MRI radiomics model based on the portal vein phase was 0.750, which was superior to that of the T2WI model (AUC: 0.710) and the arterial-phase model (AUC: 0.717). The radiomics model based on multiparametric MRI radiomics features presented the best performance (AUC: 0.831) among all radiomics models in the training cohort. The combined nomogram (AUC: 0.873) incorporated clinicopathologic radiologic (CPR) factors (MVI, pathological grading, and tumor size) outperformed the radiomics model and the CPR model. These results described above may be related to the following factors [41, 42]: (1) MVI is one of the important prognostic factors and correlated closely with early recurrence of HCC after surgical treatment. (2) Due to the large signal differences caused by the washout of the typical enhancement method of HCC, the tumour boundary on the portal vein phase in the dynamic enhanced image can be clear so that the preoperative portal vein phase image can highlight the heterogeneity of HCC. (3) Multiparametric MRI contains more potential tumor heterogeneity information.

Reviewer 2 Report

Your paper is well written and demonstrates a fairly good prediction of PD-L2expression of HCC by utilizing MRI radiomics.  It was not clear why you selected PD-L2 instead of PD-L1 since PD-L1 inhibitors have been clinically used. PD-L2 inhibition may not be useful due to developing autoimmune encephaomyelitis. It also would be much better to show radiomics features or findings of HCC on T2 and arterial phase contrasted MR images of high and low PD-L2 expression for comparative purpose. There is no significant difference of portal vein invasion between two groups of your study. Please show T2 and arterial contrasted MR images in Fig. 2.

Author Response

Re: Dear Reviewer, thank you all for handling the review of our manuscript. Based on your highly constructive comments and suggestions, we have addressed all of your comments and suggestions. We revised the Introduction section and added the reason for we selecting PD-L2.

The modifications are as follows:

Some patients with negative PD-L1 expression still respond to PD-1 checkpoint inhibitors, suggesting that other PD-1 ligands may underlie the clinical response to these treatments [8]. PD-L2 is another ligand of PD-1that can be induced in tumors by interferon exposure, leading to immune evasion [9](PMID: 28494868). PD-L2 expression mostly associated with PD-L1 expression, can also occur in the absence of PD-L1(PMID: 28619999). Previous studies showed that PD-L2 is expressed in pancreatic ductal adenocarcinoma, non-small-cell lung cancer, glioma, and HCC and high PD-L2 expression is associated with poor prognosis [10-14]PMID: 30891423; PMID: 28619999. A study reported that PD-L2 expression is upregulated on tumor-associated macrophages (TAM), and its inhibitory effects become evident when PD-L1 function is abrogated by anti-PD-L1 monoclonal antibody (mAb). (PMID: 30357491.) . Those studies suggested that,although PD-L1/PD-1-dependent suppression is the primary mechanism of immune evasion in cancer, alternative mechanisms of PD-L2 upregulation, may compensate once PD-L1 function is dampened (PMID: 30357491); PD-L2 expression may provide information beyond that of PD-L1 in predicting clinical results to targeted immunotherapy, and is a promising marker to anti-PD-1 targeted agents [10, 11, 15] (PMID: 28619999.).

In addition, we showed T2 and contrasted MR images in Fig. 2. and replaced Fig. 2.

Round 2

Reviewer 2 Report

You made a fairly good revision to improve your paper.